# BRIDGEBENCH: AN OFFLINE CONSTRAINED MULTI-AGENT REINFORCEMENT LEARNING BENCHMARK FOR INFRASTRUCTURE MANAGEMENT

## ABSTRACT

Effective infrastructure management, particularly bridge maintenance, is critical for public safety and economic benefits. Reinforcement Learning (RL) offers a promising paradigm for optimizing maintenance policies. Real-world applications often involve multiple decision-makers, rely on pre-collected offline data, and necessitate strict adherence to operational constraints. Existing RL benchmarks and methodologies frequently fall short in simultaneously addressing these multi-agent, offline, and constrained aspects within a practical domain. To bridge this gap, we introduce BridgeBench, a novel offline constrained multi-agent RL benchmark for bridge maintenance. It provides a realistic and challenging environment for evaluating algorithms designed for complex infrastructure management tasks. We integrate various state-of-the-art single-agent and multi-agent offline constrained RL algorithms on this platform, providing insights into their performance and limitations. Our work aims to accelerate research in applying advanced RL techniques to critical real-world infrastructure challenges, fostering the development of more robust, safe, and cost-effective maintenance strategies.

## 1 INTRODUCTION

The maintenance and management of critical infrastructure, such as bridges, are paramount for public safety, economic stability, and the longevity of transportation networks (Wu et al., 2021a). As infrastructure ages globally, the challenge of efficiently allocating limited resources for inspection, repair, and replacement becomes increasingly complex (Orcesi & Frangopol, 2011; Kim & Frangopol, 2018). Suboptimal maintenance strategies can lead to catastrophic failures and significant economic disruption, especially given persistent budget limitations (Bukhsh et al., 2020; Wu et al., 2021b), underscoring the urgent need for advanced decision-making tools.

Traditional approaches to infrastructure management often rely on heuristic rules or simplified optimization models, which may struggle to adapt to dynamic conditions or handle large-scale systems. For example, classical optimization methods like dynamic programming face scalability issues when applied to large networks (Kuhn, 2010; Medury & Madanat, 2014). Heuristic algorithms, while efficient, often result in problem-specific models that lack universal applicability (Shen et al., 2023). Similarly, multi-criteria decision-making (MCDM) models are limited by their strong subjectivity, as they depend heavily on expert knowledge (Tan et al., 2021). These limitations highlight a critical gap and motivate the exploration of more robust and data-driven approaches for infrastructure management. Reinforcement Learning (RL) has emerged as a powerful paradigm for sequential decision-making, offering the potential to learn optimal policies directly from data, thereby mitigating decision subjectivity and accumulating valuable experience for long-term benefits (You et al., 2019; Khalatbarisoltani et al., 2019).

Recent studies have successfully applied RL to optimize maintenance strategies for infrastructure networks, with some works framing the problem from a centralized perspective (Lei et al., 2022; Chen et al., 2024) and others from a decentralized, multi-agent viewpoint (Zhang et al., 2024; Zhou et al., 2022). However, these pioneering efforts primarily rely on online learning, which assumes an agent can safely and freely interact with the environment. Applying RL to real-world infrastructure management under this assumption presents significant hurdles that existing benchmarks and

methodologies rarely address simultaneously. **Specifically, three critical aspects are inherent in complex real-world problems**: **offline learning**, where direct online experimentation with critical infrastructure is infeasible, costly, or unsafe, necessitating learning from pre-existing static datasets; **hard constraints**, where maintenance decisions are invariably subject to strict operational constraints and finite budgets that cannot be violated; and **multi-agent coordination**, where a network of assets may require coordinated decision-making to manage shared resources and system-level objectives. Additionally, infrastructure networks present unique characteristics where individual assets can be managed either collectively through coordinated decision-making or independently through decomposed budget allocation strategies.

Despite significant advancements in offline reinforcement learning (ORL) (Fu et al., 2020; Gulcehre et al., 2021) and constrained reinforcement learning (CRL) (Ray et al., 2019; Ji et al., 2023; Liu et al., 2024), there is a notable absence of benchmarks that integrate these dimensions within a realistic, high-stakes domain involving complex network structures. Similarly, the emerging field of offline multi-agent reinforcement learning (MARL) has produced valuable benchmarks (Formanek et al., 2023), but these often focus on coordination tasks without incorporating the hard, system-level budgetary constraints that are central to our problem. Existing platforms in the RL and infrastructure communities often fall short because they are evaluated in domains like robotics that do not capture the resource competition dynamics of a large-scale asset network, focus on online interaction, lack hard budgetary constraints, or are designed for scenarios that do not capture the unique challenges of infrastructure management.

To bridge this critical gap, we introduce **a novel benchmark for Offline Constrained Multi-agent Reinforcement Learning** tailored for infrastructure management, specifically focusing on bridge maintenance networks. It is constructed using the National Bridge Inventory (NBI) dataset, a comprehensive, real-world repository of bridge inspection and maintenance records from the U.S. Department of Transportation (Contreras-Nieto et al., 2019; Bu et al., 2015). NBI is uniquely suited as a foundation due to its vast scale, containing decades of longitudinal data, and its richness, detailing structural ratings, traffic loads, and historical maintenance actions for thousands of assets.

However, transforming raw historical records into a functional RL benchmark requires a principled modeling pipeline. First, the core of the problem is formalized as a **Markov Decision Process (MDP)**, which **abstracts raw observations into a structured state-action space and designing a utility function to navigate the trade-off between structural integrity and cost**. To enable the evaluation of long-term policies offline, a **data-driven world model is constructed to simulate the environment's stochastic dynamics**. Finally, to embed the problem within a realistic multi-agent and constrained setting, we **partitioned the assets into regional networks and inferred their operational budget constraints** from historical expenditure data. This frames the challenge as a system of agents competing for limited resources, mirroring the complexities of real-world infrastructure management.

Our work reveals that while optimization-based RL, such as `Multitask CPQ`, can discover novel and highly effective policies (achieving a **38.03%** health gain over historical data), it also highlights the central challenge for real-world deployment: reconciling this immense potential with strict operational constraints. We establish that imitation learning provides a robust safety baseline (**4.77%** gain with high fidelity), serving as a pragmatic starting point. Ultimately, this benchmark quantifies the crucial trade-off between policy optimization and constraint adherence, thereby motivating and enabling the development of a new generation of constraint-aware RL algorithms poised to unlock superior performance in safety-critical systems.

## 2 RELATED WORK

Our work is positioned at the intersection of infrastructure management, offline reinforcement learning, and multi-agent systems. This section reviews the most relevant literature and situates our contribution within this context.

**Reinforcement Learning for Bridge Maintenance** The optimization of bridge maintenance strategies represents a critical sequential decision-making problem, traditionally modeled using Markov Decision Processes (MDPs) (Matos et al., 2019; Anwar et al., 2020; Tao et al., 2021). While classical methods like dynamic programming are effective for single assets, they face significant scalability

challenges at the network level, especially under strict budget constraints (Sutton & Barto, 2018). This limitation has motivated the exploration of Reinforcement Learning (RL) as a more adaptive and scalable paradigm. A growing body of research demonstrates the potential of RL in this domain, with some studies framing the problem under a single, centralized agent (Lei et al., 2022; Chen et al., 2024) and others adopting a decentralized multi-agent (MARL) perspective (Zhang et al., 2024; Zhou et al., 2022). However, a critical barrier to real-world deployment remains: these pioneering works predominantly rely on online learning, which assumes an agent can safely and freely interact with the environment—an assumption that is untenable for high-stakes infrastructure management.

**Positioning within RL Benchmarks** To situate our contribution within the broader RL landscape, we compare NBI-Benchmark against several influential benchmarks in offline and safe RL. As shown in Table 1, existing platforms, while foundational for the community, each lack at least one of the critical dimensions required for our target problem.

Table 1: Comparison of Offline and Constrained RL Datasets/Benchmarks.

| Feature / Dataset | BridgeBench (Ours) | DSRL | Safety-Gym | D4RL |
|---|---|---|---|---|
| Offline Dataset | ✔ | ✔ | ✗ | ✔ |
| Multi-Agent Support | ✔ | ✗ | ✗ | ✗ |
| Real-World Data | ✔ | Hybrid | ✗ | ✗ |
| Native Constraint Labels | ✔ | ✔ | ✔ | ✗ |

For example, D4RL (Fu et al., 2020), a cornerstone of offline RL research, lacks native cost or constraint labels, making it unsuitable for studying constrained optimization. The DSRL benchmark (Ji et al., 2023) improves on this by providing offline datasets with explicit constraint information. However, it is designed for single-agent scenarios and is based on hybrid or simulated robotics data. Conversely, Safety-Gym (Ray et al., 2019) is built for evaluating constrained agents but operates in an online setting. A common thread is that these influential benchmarks are grounded in domains like robotics or games, which do not capture the unique dynamics of real-world infrastructure deterioration and resource competition. Our NBI-Benchmark is the first to integrate all four features, providing a unique and necessary testbed for developing practical algorithms for large-scale, real-world systems.

**Benchmarks in Other Infrastructure Domains** Within the broader infrastructure space, several valuable tools have emerged. General-purpose simulation frameworks like InfraLib (Thangeda et al., 2024) provide tools to generate synthetic data from user-defined models. In other sectors, CityLearn (Vázquez-Canteli et al., 2019) provides a popular environment for building energy management, Grid2Op (Marot et al., 2020) offers a sophisticated platform for power grid operations, and ContainerGym (Friedrich et al., 2023) has been introduced for industrial resource allocation. However, these invaluable platforms differ from our work in two fundamental ways: 1) they are primarily designed for online interaction or generate data from a simulator, rather than providing a large-scale, pre-collected dataset of historical observations; and 2) their constraints often focus on operational goals (e.g., maintaining grid stability or adhering to packing logic) rather than the hard, long-term budgetary limits common in civil engineering. To the best of our knowledge, no publicly available benchmark for network-level bridge maintenance is specifically structured for offline, constrained, multi-agent reinforcement learning. Our work directly addresses this multifaceted gap.

## 3 PROBLEM FORMULATION AND BENCHMARK DESIGN

This section establishes the study's theoretical and practical foundations. First, we model the problem as a Multi-Agent Constrained Markov Decision Process (MA-CMDP). Second, we construct a large-scale benchmark from National Bridge Inventory (NBI) data for training and evaluating offline RL agents within this framework.

### 3.1 Multi-Agent Constrained MDP Formulation

We model the bridge maintenance problem as a Multi-Agent Constrained Markov Decision Process (MA-CMDP), defined by the tuple $(\mathcal{N}, \mathcal{S}, \mathcal{A}, P, R, C, \gamma, \mathbf{W}, B)$. The agents $\mathcal{N}$ is a set of $N$ agents where each agent $i \in \mathcal{N}$ corresponds to a bridge in the network. The global state space $\mathcal{S}$ has $\mathbf{s}_t = [s_t^1, s_t^2, \ldots, s_t^N, b_t]$ consisting of individual bridge states $s_t^i$ and remaining budget $b_t$, where each bridge state $s_t^i$ includes structural rating, bridge age, physical attributes (length, width, structural type), traffic load, and environmental factors derived from NBI records. Each agent selects from the action space $\mathcal{A}$ with four discrete maintenance actions: $a = 0$ (No Action), $a = 1$ (Minor Repair), $a = 2$ (Major Repair), and $a = 3$ (Replacement), forming joint action $\mathbf{a}_t = [a_t^1, a_t^2, \ldots, a_t^N]$. The transition function $P(\mathbf{s}_{t+1}|\mathbf{s}_t, \mathbf{a}_t)$ captures stochastic bridge deterioration and maintenance outcomes, estimated from historical NBI data transitions.

The reward function $R(\mathbf{s}_t, \mathbf{a}_t, \mathbf{s}_{t+1})$ provides a scalar feedback signal that balances immediate health improvements against maintenance costs. The cost function $C(\mathbf{s}_t, \mathbf{a}_t)$ represents the total monetary cost of joint maintenance actions, derived from NBI cost records. The connectivity matrix $\mathbf{W} \in \mathbb{R}^{N \times N}$ captures spatial relationships between bridges based on geographical proximity. The budget constraint $B$ is the total budget limit that constrains cumulative maintenance expenditure.

The optimization objective seeks a policy $\pi : \mathcal{S} \to \mathcal{A}$ that maximizes expected cumulative reward subject to budget constraints:

$$\max_{\pi} \mathbb{E}_{\pi} \left[ \sum_{t=0}^{T} \gamma^t R(\mathbf{s}_t, \mathbf{a}_t, \mathbf{s}_{t+1}) \right] \quad \text{s.t.} \quad \mathbb{E}_{\pi} \left[ \sum_{t=0}^{T} C(\mathbf{s}_t, \mathbf{a}_t) \right] \leq B$$

This formulation supports both multi-agent approaches for resource competition and coordination, and single-agent approaches where budget allocation decomposes the network problem into individual subproblems. The multi-agent view suits the distributed nature of infrastructure management, where decision-makers compete for limited resources. Due to safety and cost, direct experimentation on critical infrastructure is infeasible, necessitating offline learning. Hard budget constraints reflect strict public sector budgets where overspending is intolerable.

### 3.2 Benchmark Construction with NBI Data

The construction of our benchmark dataset follows a structured pipeline, as illustrated in Figure 1, designed to transform raw National Bridge Inventory (NBI) records into a format suitable for reinforcement learning. We utilize highway bridge data from California spanning 1992 to 2023 to ensure data homogeneity and sufficient sample size.

**Data Preprocessing and MDP Formulation.** The process begins with raw data extraction and a rigorous preprocessing stage. We then formulate the core components of the Markov Decision Process (MDP). The bridge's health state is categorized into four distinct levels based on structural evaluation scores: *Good* (rating $\geq 7$), *Fair* ($5 \leq$ rating $< 7$), *Poor* ($3 \leq$ rating $< 5$), and *Critical* (rating $< 3$). The action space is defined by mapping NBI's work codes to four categories: No Action, Minor Repair, Major Repair, and Replacement, with empirically derived average costs of \$0, \$71.56, \$1,643.31,

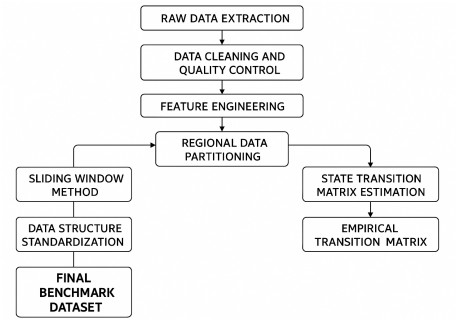

Figure 1: The pipeline for constructing the benchmark dataset from raw NBI data. The process includes data preprocessing, MDP formulation, regional partitioning, and yields the final dataset and an empirical transition matrix.

and \$2,433.53, respectively. The reward function is designed to balance health improvement against cost-effectiveness:

$$R(s_t, a_t, s_{t+1}) = \Delta \text{Health} - \beta \cdot \text{cost}(a_t) + \text{Bonus/Penalty}$$

where $\Delta$Health represents the change in health state. The function penalizes action costs, weighted by a parameter $\beta$, and includes bonus or penalty terms to encourage proactive maintenance on high-health bridges and discourage neglect of critical ones. The detailed construction of these terms and the parameter selection for $\beta$ are provided in Appendix B.

**Multi-Agent Environment and Dataset Construction.** To create meaningful multi-agent scenarios, we partition the statewide dataset into 400 geographically coherent regions using a spatial clustering algorithm. For each region, a static connectivity matrix $\mathbf{W}$ is constructed to capture inter-bridge relationships. From the time-series data of each region, we generate multiple episodes using a sliding window approach. This process culminates in a benchmark dataset of 2000 episodes. Each episode is stored as a collection of NumPy arrays, containing observations (with a feature dimension of 5), actions, and rewards for all agents over a 25-step horizon, alongside the region's static connectivity matrix. Finally, global normalization is applied across all features. The specifics of the regional partitioning, dataset generation, and final data structure are further elaborated in Appendix B.

**Empirical Transition Matrix Estimation.** In parallel with the main dataset construction, we estimate an empirical state transition matrix, $P(s_{t+1}|s_t, a_t)$, by aggregating observed year-over-year health state changes for each action type across the entire dataset. This matrix quantifies the average system dynamics, providing a valuable resource for model-based reinforcement learning approaches. The detailed methodology for its construction and the resulting transition matrices are presented in Appendix B.

## 4 EVALUATED ALGORITHMS AND EXPERIMENTAL SETUP

Our reliance on a fixed historical dataset necessitates an offline reinforcement learning approach. The evaluated algorithms were specifically chosen to explore solutions along two primary axes of our benchmark's challenges: the management of explicit budget constraints and the decentralized, multi-agent nature of the problem. Details for each algorithm are provided in Appendix A.

### 4.1 ALGORITHM OVERVIEW

To address the challenges outlined above, our selected algorithms are categorized into two main groups. First, we include single-agent algorithms that explicitly handle constraints. These methods, such as the value-based `multitask_CPQ` and the imitation-learning-based `CDT` and `multitask_bc`, treat the problem as a centralized decision-making task under strict budgetary limits and serve to benchmark performance when global optimization is attempted.

Second, to reflect the operational reality of distributed management, we evaluate a range of multi-agent algorithms. This includes fundamental baselines like a `Random` policy and `discrete_bc` (Behavior Cloning) to establish lower and expert-imitation performance bounds, respectively. Furthermore, we incorporate state-of-the-art offline MARL methods to assess the potential for learning policies that improve upon historical data, including `IQL-CQL` for independent learning and `QMIX-CQL` for centralized training with coordination.

Table 2 provides a systematic comparison of their key features.

### 4.2 EVALUATION METRICS

A set of evaluation metrics is designed to capture the performance and effectiveness from multiple perspectives.

**Health Improvement.** In reality, the overall health of a bridge network deteriorates annually. The goal of a maintenance strategy is to mitigate this degradation. Using the state transition matrix previously defined, we calculate the average health change per bridge under the algorithm's policy. This change is then compared against two baselines: the outcome of doing nothing (no maintenance) and the outcome following historical maintenance actions. These comparisons correspond to the "Improve vs None" and "Improve vs History" values in our results table, quantifying how much better the algorithm performs relative to these benchmarks.

Table 2: Algorithm Feature Comparison

| Algorithm | Core Features | | | Advanced/Specific Features | | |
| --- | --- | --- | --- | --- | --- | --- |
| | Explicit Constraint Handling | Single-Agent | Multi-Agent | Centralized Training (MARL) | Behavior Cloning | Value-Based |
| *Single-Agent Algorithms* | | | | | | |
| `multitask_CPQ` | ✔ | ✔ | ✗ | N/A | ✗ | ✔ |
| CDT | ✔ | ✔ | ✗ | N/A | ✔ | ✗ |
| `multitask_bc` | ✔ | ✔ | ✗ | N/A | ✔ | ✗ |
| *Multi-Agent Algorithms* | | | | | | |
| Random | ✗ | ✗ | ✔ | N/A | ✗ | ✗ |
| `discrete_bc` | ✗ | ✗ | ✔ | ✗ | ✔ | ✗ |
| IQL-CQL | ✗ | ✗ | ✔ | ✗ | ✗ | ✔ |
| QMIX-CQL | ✗ | ✗ | ✔ | ✔ | ✗ | ✔ |

**Budget Ratio.** As the name suggests, this metric is the ratio of the total cost incurred by the algorithm to the total available budget, indicating the proportion of the budget that was utilized.

**Behavioral Similarity.** This metric measures the proportion of an algorithm's actions that are identical to the actions taken in the historical data, reflecting the degree to which the algorithm mimics past human expert decisions.

**Violation Rate.** This is the proportion of individual actions for which the cost exceeds the allocated budget for that specific decision. It measures the algorithm's ability to adhere to financial constraints on a per-action basis.

**Health Gain per \$1M.** This metric evaluates the cost-efficiency. It is calculated as the average health improvement per bridge, divided by the average cost per bridge, and then scaled by \$1,000,000. It represents the health gain achieved for every million dollars invested. In our evaluation tables, a higher value for "Health Gain per \$1M" signifies a more efficient.

## 5 Results and Analysis

This section presents a multi-faceted evaluation of the trained algorithms, designed to assess their effectiveness, efficiency, and robustness. Our experimental framework is extensive: we evaluate **8 core algorithms** under two primary evaluation paradigms. The first is a static assessment on a held-out test set. The second is a dynamic, 100-year longitudinal simulation that further examines algorithm behavior across **5 distinct budget scaling factors** ($0.25\times$ to $4.0\times$) and **9 different budget allocation strategies**. To ensure statistical robustness, each unique experimental configuration was repeated with multiple random seeds.

### 5.1 Quantitative Performance on the Test Set

To establish a baseline understanding of each algorithm's capabilities, we first evaluate their performance on a held-out test set. This static evaluation is crucial for revealing the inherent trade-offs between maximizing health outcomes and adhering to budgetary constraints, as captured by the metrics in Table 3 and visualized in Appendix C.

The results reveal a critical constraint-performance paradox: algorithms achieving the highest performance gains consistently exhibit the poorest constraint adherence. **Multitask CPQ** dominates the upper-left corner with exceptional health improvements (54.02% over no-action, 38.03% over historical baseline) and remarkable efficiency (12.88 health units per \$1M), yet suffers from a severe 71.8% violation rate that renders it impractical for deployment.

In contrast, **discrete BC** and **IQL-CQL** algorithms demonstrate different compromises. `discrete_bc_50` achieves a competitive health improvement (4.77% over historical baseline) with a reasonable budget ratio ($1.018\times$) and the highest behavioral similarity score (0.945).

Table 3: Algorithm Performance Comparison

| Algorithm | Budget Ratio | Improve vs None | Improve vs History | Behavioral Similarity | Violation Rate | Health Gain per $1M |
|---|---|---|---|---|---|---|
| *Single-Agent Algorithms* | | | | | | |
| multitask_bc | $1.638 \pm 0.09$ | $-5.16 \pm 4.24$ | $-41.52 \pm 5.70$ | $0.673 \pm 0.01$ | $0.020 \pm 0.00$ | -1.1446 |
| cdt | $0.419 \pm 0.06$ | $19.81 \pm 3.42$ | $-7.23 \pm 4.60$ | $0.851 \pm 0.01$ | $0.064 \pm 0.01$ | 5.2475 |
| multitask_cpq | $\mathbf{0.283 \pm 0.05}$ | $\mathbf{54.02 \pm 0.91}$ | $\mathbf{38.03 \pm 1.22}$ | $0.146 \pm 0.01$ | $0.718 \pm 0.00$ | **12.8769** |
| *Multi-Agent Algorithms* | | | | | | |
| iqlcql_marl_no_bud | $0.136 \pm 0.04$ | $13.23 \pm 4.43$ | $-16.84 \pm 5.96$ | $0.868 \pm 0.00$ | $0.014 \pm 0.00$ | 1.7547 |
| iqlcql_marl | $0.897 \pm 0.07$ | $23.57 \pm 2.62$ | $-2.95 \pm 3.52$ | $\mathbf{0.942 \pm 0.00}$ | $0.086 \pm 0.00$ | 6.4346 |
| discrete_bc_50 | $1.018 \pm 0.04$ | $29.31 \pm 2.60$ | $4.77 \pm 3.49$ | $\mathbf{0.945 \pm 0.00}$ | $0.087 \pm 0.00$ | 7.7829 |
| qmix_cql | $1.625 \pm 0.15$ | $20.91 \pm 6.59$ | $-6.52 \pm 8.87$ | $0.675 \pm 0.02$ | $0.232 \pm 0.01$ | 3.2430 |
| random_marl | $4.112 \pm 0.63$ | $4.23 \pm 0.61$ | $-28.91 \pm 0.82$ | $0.249 \pm 0.00$ | $0.539 \pm 0.00$ | 0.8531 |

iqlcql_marl, while showing a slight performance degradation against the historical baseline (-2.95%), maintains a strict budget ratio (0.897×) and high behavioral similarity (0.942). These results indicate that imitation-based methods can strongly align with expert decision patterns while maintaining moderate violation rates (8.6–8.7%).

Multi-agent algorithms generally achieve superior behavioral similarity, with discrete BC variants and IQL-CQL showing the highest scores, indicating better capture of expert decision patterns. The efficiency ranking clearly emerges as: CPQ ≫ multi-agent BC > IQL-CQL > CDT > QMIX-CQL ≫ random > zero improvement > single-agent BC.

## 5.2 ACTION SELECTION ANALYSIS

For a deeper insight into algorithmic behavior, we analyze action selection under two different evaluation settings. It is important that during training, we do not apply any explicit constraints on the agents' action outputs; the algorithms learn purely from the provided data. For the evaluation, however, we compare two scenarios: an **unrestricted** setting, where agents freely choose actions based on their learned policies, and a **budget-restricted** setting, where we enforce a hard constraint. In this latter case, shown on the right side of Figure 2, agents are only permitted to select actions with a cost less than the remaining budget. This comparison reveals how budget constraints fundamentally alter behavior and provides insights into the mechanisms behind constraint adherence.

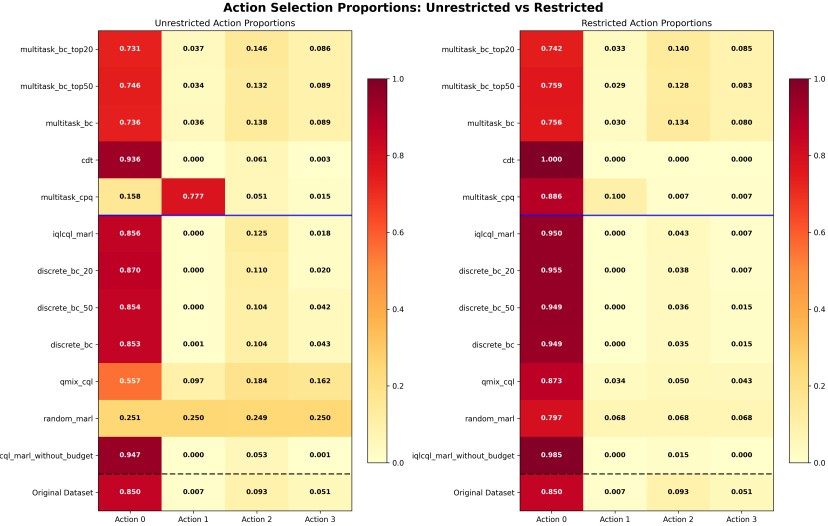

Figure 2: Action distribution under unrestricted (left) and budget-restricted (right) conditions. Most algorithms align with the dataset's bias toward Action 0 (85.0%), except CPQ which shows a learned preference for Action 1.

Most algorithms align closely with the historical dataset's bias toward **Action 0** (85.0%), with multi-agent BC algorithms exhibiting nearly identical action distributions. The notable exception is **multitask CPQ**, which demonstrates a learned preference for low-cost **Action 1** (77.7%) under unrestricted conditions, diverging significantly from dataset patterns. This preference for Action 1 may precisely explain CPQ's exceptional efficiency, as it discovered a cost-effective strategy underutilized in the original dataset.

A clear pattern emerges: algorithms with higher violation rates exhibit greater behavioral changes when budget constraints are imposed. CPQ and QMIX-CQL, which have high violation rates, show dramatic shifts in action selection under restrictions. Conversely, single-agent BC algorithms demonstrate minimal changes between unrestricted and restricted conditions, indicating effective constraint adherence and consistent policy behavior.

### 5.3 100-YEAR LONG-TERM SIMULATION ANALYSIS

While static evaluation provides a snapshot of performance, critical questions remain: How do these policies perform over an extended operational horizon? How sensitive are they to changes in budget availability and allocation? To answer these, we designed a 100-year longitudinal simulation. The simulation is initialized using data from a specific geographic region in our test set, with the crucial modification that all bridges are set to their maximum health state. This allows us to observe the full life-cycle performance. The simulation progresses annually, with bridge health evolving according to our pre-defined state transition matrix. For years extending beyond the original dataset's timeframe, a bridge's age increments naturally and its health state changes, while other static observational features are held constant.

**Long-term Performance Comparison.** To establish a baseline for long-term behavior, we first analyze performance under the **original historical budget allocation strategy** and a standard **1.0× budget ratio**. In this default setting, we observe substantial disparities over the 100-year simulation (Figure 3). `multitask_cpq` maintains its exceptional performance, achieving both the highest health gains and superior cost efficiency. In contrast, moderate performers on the test set like `cdt` and `qmix_cql` exhibit high expenditure with poor cost efficiency. This counterintuitive pattern primarily stems from the distribution shift between the simulation environment (which starts with healthy bridges) and the training data, challenging the algorithms' ability to generalize to out-of-distribution states.

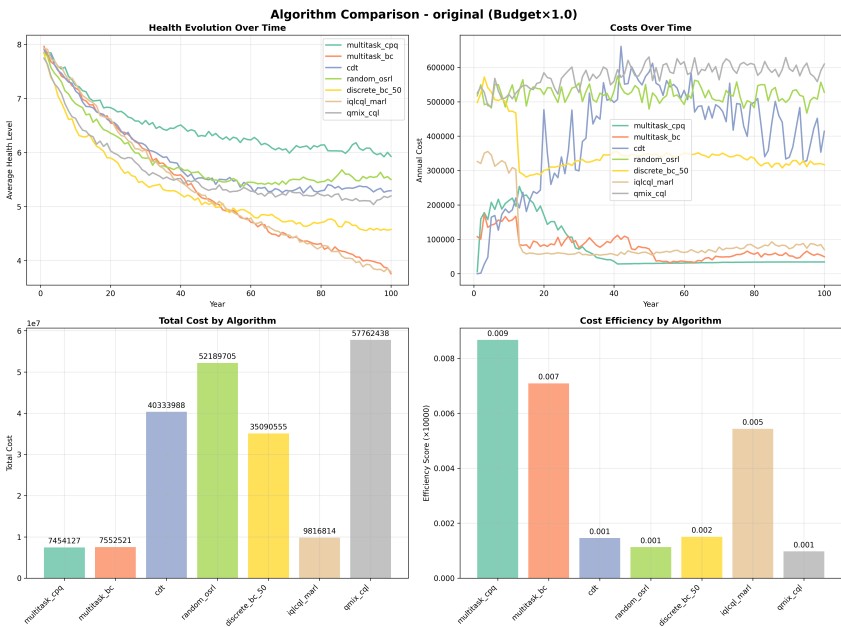

Figure 3: 100-year simulation comparison under original budget allocation (1.0× scaling). Shows health evolution, budget expenditure, total costs, and efficiency metrics across algorithms.

**Budget Sensitivity Analysis.** Examining algorithm response to five budget scaling factors ($0.25\times$ to $4.0\times$) reveals two distinct behavioral categories (Figure 4). *Budget-sensitive* algorithms (e.g., `discrete_bc`, `iqlcql_marl`) immediately select higher-cost actions when constraints are relaxed, showing a learned association between budget and action choice. However, their efficiency often decreases due to diminishing returns. *Budget-insensitive* algorithms (e.g., `cdt`, `multitask_cpq`) exhibit more irregular patterns, as their optimization objectives (like achieving a target return) may not prioritize spending the entire budget.

**Impact of Budget Allocation Strategy.** Our analysis of nine different allocation strategies reveals remarkable consistency: the **original dataset's allocation strategy** emerges as a well-balanced approach. While more targeted strategies (e.g., uniform top-10% allocation) achieve higher efficiency through lower expenditure, the reduced investment leads to suboptimal long-term health outcomes. This validates the underlying rationality of the historical resource distribution patterns. Detailed strategy comparisons are provided in Appendix C.

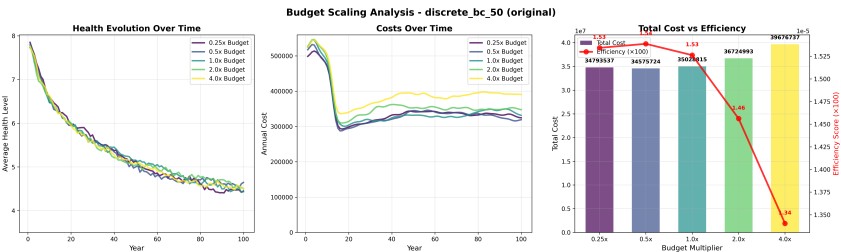

Figure 4: Budget scaling sensitivity for discrete BC (representative of budget-sensitive algorithms). Other algorithms show varying degrees of sensitivity, with detailed analysis in Appendix C.

## 6 DISCUSSION AND CONCLUSION

This work establishes the first comprehensive benchmark for multi-agent offline constrained reinforcement learning in large-scale infrastructure maintenance. Our findings reveal a fundamental "constraint-performance paradox" that highlights a critical gap between the theoretical potential of optimization algorithms and the pragmatic demands of real-world deployment. Algorithms that achieve the highest performance gains do so by violating operational constraints, while those that adhere to constraints offer more modest, yet reliable, improvements.

A central insight is that optimization-based methods like `multitask_CPQ` can discover novel, highly efficient maintenance strategies. Specifically, its preference for frequent, low-cost preventative actions (a 38.03% health improvement) demonstrates RL's capacity to uncover policies that diverge from and potentially improve upon established human practice. However, this unconstrained optimization leads to a 71.8% budget violation rate, rendering the policy operationally infeasible. In contrast, imitation-based methods such as `discrete_bc` provide a pragmatic and safe path for near-term deployment. By closely mimicking expert decisions, they not only achieve a respectable 4.77% health gain but also inherit the implicit risk management and budget adherence of the historical data, making them a reliable starting point.

Our 100-year simulations further underscore these challenges, revealing that even effective policies may lead to unsustainable expenditure over a century-long horizon. This is compounded by distribution shift, where policies falter when faced with conditions not seen in the training data, such as a network of initially healthy bridges. Ultimately, BridgeBench formalizes the trade-off between pure optimization and practical, constraint-bound deployment. The path forward requires moving beyond both unconstrained optimization and pure imitation. This benchmark thus serves as both a foundation and a challenge to the research community: to develop a new generation of algorithms that are inherently constraint-aware, robust to long-term dynamics, and can reconcile the discovery of novel, high-performance policies with the strict operational realities of the real world. More detailed discussion is provided in Appendix E.

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
