# Supplementary Material

## Table of Contents

## A   DETAILED ALGORITHM DESCRIPTIONS

This appendix provides detailed descriptions of the offline reinforcement learning algorithms evaluated in our benchmark. Each description outlines the core mechanism, architecture, and training objective of the respective algorithm.

### A.1   MULTI-AGENT OFFLINE ALGORITHMS

Our benchmark evaluates several multi-agent offline reinforcement learning algorithms. These methods are designed to learn decentralized policies from pre-collected, static datasets without further interaction with the environment, which is crucial for real-world applications where online exploration is infeasible. For the multi-agent algorithm, we refer to the implementation in og-marl(Formanek et al., 2023)

**Random.** The random baseline serves as a fundamental lower bound for multi-agent performance. In this setup, each agent independently and randomly selects an action from the available space at each step. This policy represents a complete lack of coordination or learned behavior. A specialized

version can incorporate rudimentary budget awareness by prioritizing "affordable" actions, but it remains a non-learning-based approach.

**discrete_bc.** This algorithm is a multi-agent adaptation of Behavior Cloning (BC) for discrete action spaces. It learns a decentralized policy by directly mimicking expert actions from the offline dataset. Each agent's policy is represented by a `DeepRNN` (Recurrent Neural Network), which processes the agent's local observation. To enable agents to distinguish themselves, a one-hot encoded agent ID is concatenated to each agent's observation. The training objective is to minimize the cross-entropy loss between the predicted action probabilities and the expert's chosen actions. This algorithm serves as a strong imitation learning baseline.

**IQL-CQL.** The `IQL-CQL` algorithm extends the Individual Q-learning (IQL) framework with Conservative Q-Learning (CQL) for multi-agent settings. Each agent learns its own Q-function using a `DeepRNN`, where agent IDs are also appended to observations. The core idea of IQL is to train each agent's Q-function independently. CQL is integrated to address the overestimation bias inherent in offline Q-learning by adding a regularization term to the objective that penalizes Q-values for out-of-distribution actions. The overall loss combines the standard TD (Temporal Difference) error with the CQL regularization term.

**QMIX-CQL.** `QMIX-CQL` combines the centralized training with decentralized execution (CTDE) framework of QMIX with CQL for offline learning. Individual agents learn their own Q-functions (using `DeepRNNs` with agent ID concatenation), but their Q-values are combined by a monotonic mixing network (`QMixer`) to produce a global Q-value. For offline learning, CQL regularization is applied to the mixed global Q-values, pushing down the values of actions not present in the dataset. This approach leverages multi-agent coordination through the mixer while ensuring conservative Q-value estimates for reliable offline policy learning.

## A.2 CONSTRAINED SINGLE-AGENT OFFLINE ALGORITHMS

To provide a contrasting perspective to the decentralized multi-agent approaches, we also evaluate several single-agent algorithms. These methods treat the entire system as a centralized control problem, where a single policy makes all decisions. This allows us to benchmark the performance of global optimization strategies, particularly those designed to handle explicit constraints. For the single-agent algorithm, we refer to the implementation in osrl(Liu et al., 2024)

**multitask_bc.** This is a single-agent Behavior Cloning (BC) approach that learns a policy by mimicking expert demonstrations from the entire dataset. It is tailored for discrete action spaces and handles dynamic budget information by concatenating the current `budget` with the `observation` as input to the actor network (`MLPActorDiscrete`). This allows the policy to condition its actions on the remaining budget. Training is performed by minimizing the `nn.CrossEntropyLoss`.

**CDT (Constrained Decision Transformer).** CDT adapts the Decision Transformer architecture for constrained environments. It is designed to strictly adhere to constraints by conditioning its predictions on both future returns (return-to-go) and future costs (cost-to-go). The transformer predicts actions based on the trajectory context (past states, actions, and rewards), while the attention mechanism allows it to respect cumulative cost constraints. The model is trained to imitate expert trajectories from the dataset that satisfy the given budget constraints.

**multitask_CPQ (Constrained Policy Q-learning).** `multitask_CPQ` is an adaptation of Constrained Policy Q-learning (CPQ), a model-free, off-policy algorithm for constrained RL with discrete action spaces. This "budget-aware" algorithm maintains two Q-networks: one for the expected cumulative reward (`q_net`) and another for the expected cumulative cost (`qc_net`). Both networks take the concatenated `state` and `budget` as input. For constraint handling, a cost threshold (`qc_thres`) is calculated based on the cost limit. Actions predicted to lead to a future cumulative cost exceeding this threshold are pruned by setting their Q-values to a very low number, ensuring the policy learns to avoid constraint violations.

## B    DETAILED NBI DATA PROCESSING PIPELINE

Our dataset construction follows a comprehensive multi-stage pipeline designed to transform raw NBI records into a structured reinforcement learning benchmark. This is shown in Figure 1

### B.1    DATA ACQUISITION AND PREPROCESSING

Our benchmark is derived from the National Bridge Inventory (NBI), a longitudinal database maintained by the U.S. Federal Highway Administration (FHWA) that contains comprehensive records for all public road bridges in the United States.[1] Our multi-stage data processing pipeline transforms these raw records into a structured reinforcement learning benchmark.

**Data Extraction and Cleaning.** We utilized annual NBI data files from 1992 to 2023, performing an initial filtering to retain only highway bridges ('SERVICE_ON_042A' = 1) located in California. This focus on a single state and infrastructure type minimizes confounding variables related to differing state policies and environmental conditions. To address common issues in historical data, we implemented a rigorous cleaning protocol. Missing temporal data, such as structural evaluation scores, were imputed using a forward-fill followed by a backward-fill strategy to maintain the temporal consistency of each bridge's condition history. To ensure that our analysis was based on assets with sufficient historical context, bridges with fewer than 20 years of records within our study period were excluded. Furthermore, we applied statistical outlier detection methods based on interquartile range to identify and correct or cap anomalous values in cost and rating fields, which likely represent data entry errors.

**Feature Engineering.** To enrich the state representation beyond raw NBI fields, we engineered several informative features designed to capture critical aspects of a bridge's condition and importance. These include 'bridge_age' (calculated as 'current_year' - 'YEAR_BUILT_027'), which is a primary driver of deterioration. We also computed 'traffic_density' as the ratio of 'Average_Daily_Traffic_029' to 'Deck_Width_052', serving as a proxy for the operational stress on the structure. To capture the recent health trajectory, we calculated a 'deterioration_rate' as the average change in the primary structural rating over the preceding five years. Finally, we defined a composite 'importance_score' as a weighted sum of normalized ADT, age, and span length to approximate the bridge's systemic importance, helping an agent to prioritize critical assets.

**Action Space Definition and Cost Estimation.** To create a tractable action space, we abstracted the specific maintenance codes from the NBI field 'WORK_PROPOSED_075A' into four discrete, high-level actions: **No Action**, **Minor Repair**, **Major Repair**, and **Replacement**. The cost associated with each action, 'cost(a_t)', was empirically derived by calculating the average of the inflation-adjusted 'TOTAL_IMP_COST_096' across all corresponding historical interventions. The cost for 'No Action' is defined as zero. Table 4 provides a detailed summary of this mapping and the resultant cost structure used throughout our experiments.

Table 4: Mapping of NBI Work Codes to Action Categories and Associated Costs

| Action Category | Assigned NBI Work Codes | Calculated Avg. Cost ($) |
| --- | --- | --- |
| No Action | 0 | 0.00 |
| Minor Repair | 33 | 71.56 |
| Major Repair | 31, 34, 35 | 1643.31 |
| Replacement | 32, 36, 37, 38 | 2433.53 |

### B.2    NATIONAL BRIDGE INVENTORY (NBI) FEATURE DETAILS

To provide clarity on the raw data used, this section details the key NBI items referenced in our pipeline. Definitions are based on the FHWA's *Recording and Coding Guide*. The bridge's health state is categorized into four distinct levels based on structural evaluation scores: **Good** (rating $\geq$ 7), **Fair** ($5 \leq$ rating $7$), **Poor** ($3 \leq$ rating $5$), and **Critical** (rating $3$).

---

[1]The NBI public data is accessible at: https://www.fhwa.dot.gov/bridge/nbi/ascii.cfm

**Condition Ratings (Items 58, 59, 60).** The core of the state representation is derived from the condition ratings for the Deck, Superstructure, and Substructure. Each is rated on a 0-9 scale, as detailed in Table 5. Our primary health score is the minimum of these three values, representing the weakest link principle.

Table 5: NBI Condition Rating Scale and Our Health State Categorization

| Code | Description | Our Categorization |
|------|-------------|--------------------|
| 9 | EXCELLENT | Good |
| 8 | VERY GOOD | Good |
| 7 | GOOD | Good |
| 6 | SATISFACTORY | Fair |
| 5 | FAIR | Fair |
| 4 | POOR | Poor |
| 3 | SERIOUS | Poor |
| 2 | CRITICAL | Critical |
| 1 | "IMMINENT" FAILURE | Critical |
| 0 | FAILED | Critical |

**Action and Cost Items.** Our action space and costs are derived from the following fields:

- WORK_PROPOSED_075A: This field indicates the type of work proposed to be done on the bridge. Our action mapping uses the following codes: 31 (Widening), 32 (Deck Replacement), 33 (Deck Widening), 34 (Rehabilitation), 35 (Repair), 36 (Strengthening), 37 (Painting), 38 (Other).
- TOTAL_IMP_COST_096: The estimated total cost of the improvement proposed in Item 75A, recorded in thousands of dollars. We use this to derive the empirical cost for our action space.

**State and Feature Items.** The following fields are used to construct the state space and engineered features:

- SERVICE_ON_042A: Type of service on the bridge. We filter for code '1', indicating a highway bridge.
- YEAR_BUILT_027: The year the bridge was originally constructed, used to calculate 'bridge_age'.
- AVERAGE_DAILY_TRAFFIC_029: The average number of vehicles per day carried by the bridge.
- DECK_WIDTH_052: The out-to-out width of the bridge deck in meters, used to calculate 'traffic_density'.
- LAT_016 & LONG_017: The latitude and longitude of the bridge, used for our geographic regional partitioning.

B.3 REWARD FUNCTION FORMULATION

The reward function is central to guiding agent behavior and is composed of two primary components: a health-based reward $R_{\text{health}}$ and a cost-based penalty $R_{\text{cost}}$. The total reward for a single agent is $R(s_t, a_t, s_{t+1}) = R_{\text{health}} - R_{\text{cost}}$.

**Health Reward Component ($R_{\text{health}}$).** This component is designed to reflect the change in a bridge's structural integrity and incentivize proactive maintenance. Its calculation depends on whether the bridge's condition changed over the timestep. To handle potential data anomalies, any recorded single-year evaluation change greater than 6 points is disregarded. If the structural evaluation score remains unchanged, a static reward is assigned based on the current condition: a positive reward is given for maintaining a 'Good' state (score 7), a smaller positive reward for a 'Fair' state (score 5), and a penalty for remaining in a 'Critical' state (score 3). This encourages preserving good condition cost-effectively while penalizing inaction on failing assets. If the evaluation score

changes, the health reward is dynamic and directly proportional to the magnitude of this change, providing a strong signal for actions that lead to tangible improvements or deteriorations.

**Cost Penalty Component ($R_{cost}$) and Parameter Selection.** The cost penalty is defined as $R_{cost} = \beta \cdot cost(a_t)$, where $cost(a_t)$ is the empirically derived average cost for a given action and $\beta$ is a critical hyperparameter. The selection of $\beta$ was guided by an empirical analysis on over 615,000 transition samples. We performed a sensitivity analysis to find a $\beta$ value that ensures the expected cost penalty and the expected health reward are of a similar order of magnitude, preventing one component from systematically dominating the learning signal. Our goal was to find a $\beta$ that places the ratio of the average cost penalty to the average health reward magnitude within a target range of [0.5, 2.0]. The key results of this analysis are summarized in Table 6.

Table 6: Sensitivity analysis for the cost-weighting parameter $\beta$. The ratio of the average cost penalty to the average health reward magnitude is shown for different $\beta$ values. Our goal was a ratio between 0.5 and 2.0.

| $\beta$ **Value** | **Cost/Health Ratio** | **Comment** |
| --- | --- | --- |
| $1.0 \times 10^{-5}$ | 0.01 | Cost is negligible; agent would ignore costs. |
| $5.0 \times 10^{-4}$ | 0.52 | Balanced; cost is a significant consideration. |
| $1.0 \times 10^{-3}$ | **1.05** | **Well-balanced; chosen value.** |
| $1.0 \times 10^{-2}$ | 10.50 | Cost is dominant; agent would be overly frugal. |

Based on this analysis, we selected $\beta = 1.0 \times 10^{-3}$ for all experiments. As the table demonstrates, this value yields a cost-to-health reward ratio of approximately 1.05, creating a reward signal that appropriately values both structural health and economic efficiency.

## B.4 STATE TRANSITION MATRIX ESTIMATION

We constructed an empirical state transition matrix, $P(s_{t+1}|s_t, a_t)$, for each of the four action types by aggregating all observed one-year transitions in the dataset and normalizing the counts into probabilities. These matrices (visualized in Figure 5 in the main text) empirically validate our action categorization. The matrix for **No Action** shows a strong diagonal and sub-diagonal, indicating high probabilities of a bridge remaining in its current state or deteriorating. The matrix for **Minor Repair** shows a modest upward shift in probability mass, primarily serving to arrest deterioration. In contrast, the matrix for **Major Repair** offers substantial restoration, with significant probabilities of moving bridges from 'Poor' or 'Critical' states to 'Fair' or 'Good'. Finally, the **Replacement** matrix overwhelmingly concentrates probability mass in the 'Good' state column, correctly representing a complete renewal of the asset.

## B.5 REGIONAL PARTITIONING AND EPISODE GENERATION

**Neighborhood-based Regional Partitioning.** To create meaningful multi-agent scenarios that reflect localized management challenges, we partitioned the statewide dataset into geographically coherent regions. Instead of a global clustering algorithm, we employed a neighborhood-based sampling method. To form a single region, a bridge was first randomly selected from the entire pool to act as a seed. Then, the $N - 1$ bridges with the smallest Manhattan distance (calculated from their latitude and longitude coordinates) to this seed were identified. This collection of $N$ bridges—the seed and its nearest neighbors—constitutes one region. This process was repeated, sampling without replacement, until 400 distinct regions were generated. This approach ensures that agents within an episode manage a set of geographically proximate assets and face realistic, localized resource competition. For each region, a static, binary connectivity matrix $\mathbf{W}$ is constructed based on geographical proximity, capturing the inter-bridge spatial relationships.

**Sliding Window for Episode Generation.** From each region's full time-series data, we generated multiple episodes by applying a sliding window of 15 years with a stride of 5 years. This technique augments the number of distinct trajectories available for training while ensuring that each episode maintains its internal temporal coherence. This process resulted in a final benchmark dataset of 2000 episodes.

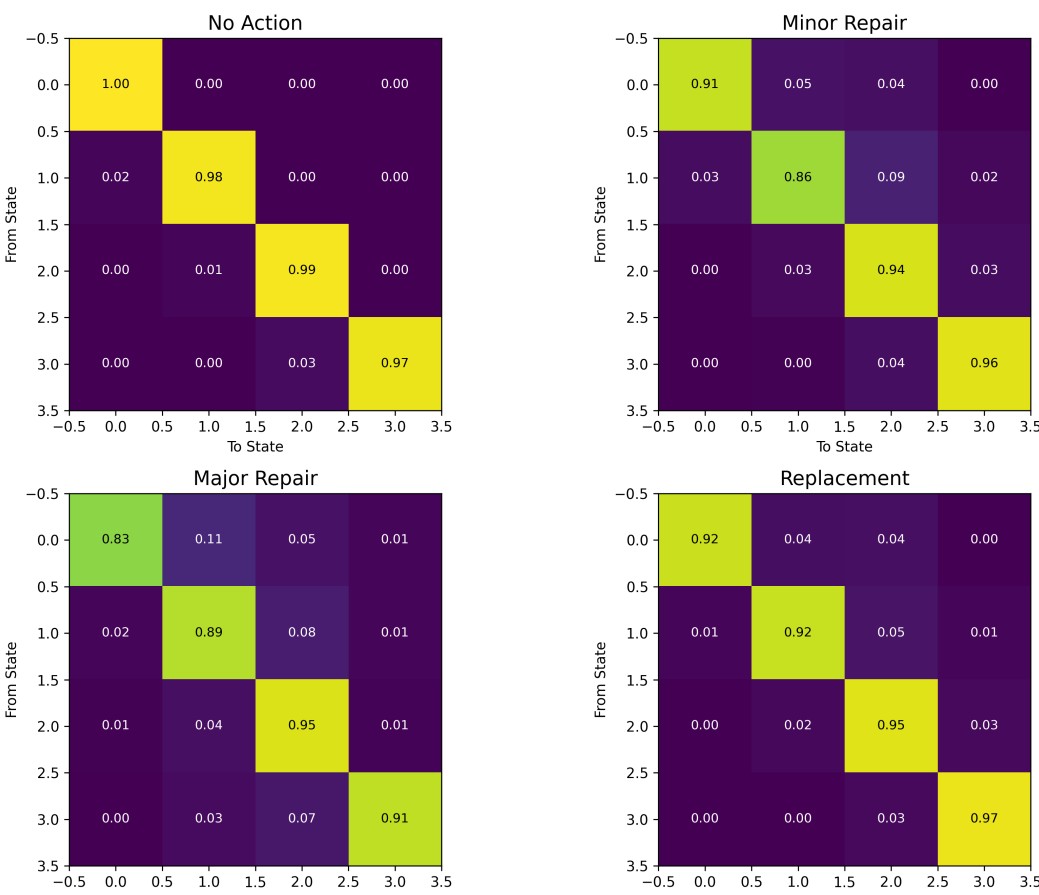

Figure 5: State transition matrices for different maintenance actions. Each matrix shows the probability of transitioning from one health state (rows) to another (columns) given a specific action. The four sub-figures correspond to: (top-left) No Action, (top-right) Minor Repair, (bottom-left) Major Repair, (bottom-right) Replacement.

### B.6 FINAL DATASET STRUCTURE AND NORMALIZATION

**Data Structure Specification.** Each episode in the final benchmark is stored as a dictionary-like object containing a collection of NumPy arrays. The dimensions below represent a single episode, with $T$ being the time horizon and $N$ the number of agents in the region. The arrays include: 'obs_arr' $[T, N, \text{obs\_dim}]$, 'act_arr' $[T, N]$, 'rew_arr' $[T, N]$, 'cost_arr' $[T, N]$, the static 'connectivity' matrix $[N, N]$, the shared 'budget_arr' $[T]$, and a 'metadata' dictionary containing episode-specific information.

**Global Normalization.** All continuous features were normalized using parameters computed solely from the training set to prevent data leakage. We employed a mixed strategy tailored to feature characteristics: **Z-score normalization** for features with Gaussian-like distributions (e.g., traffic density); **Min-max normalization** for features with defined bounds (e.g., bridge age); and **Robust scaling** (using median and interquartile range) for features with significant outliers, such as maintenance costs.

## C    MORE EXPERIMENTAL DETAILS

This section provides a more granular analysis of the experimental results presented in Section 5.1, offering deeper insights into algorithm performance, budget sensitivity, and the effectiveness of various resource allocation strategies.

### C.1    HOW IS THE ALGORITHM PERFORMANCE?

The static evaluation on the held-out test set provides a clear visualization of the fundamental trade-offs inherent in this task. Figure 6, which plots health improvement versus budget ratio, serves as the primary basis for this analysis. The ideal algorithm would reside in the upper-left quadrant, indicating a greater health improvement than the historical baseline for a lower relative cost. We analyze each algorithm's position on this plot to understand its specific strengths and weaknesses.

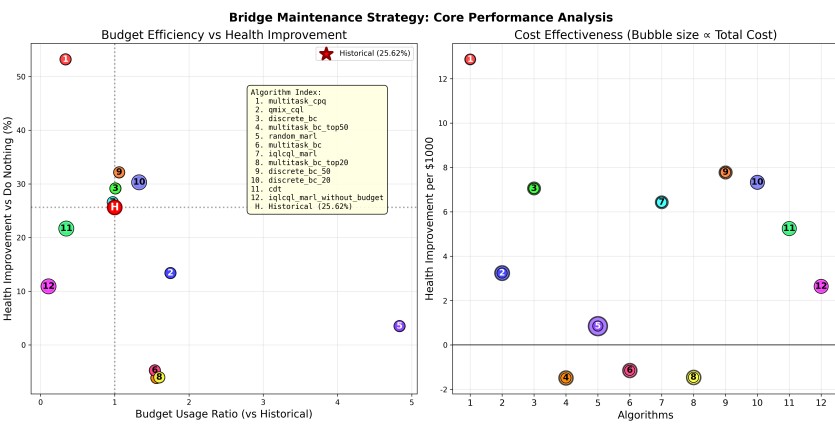

Figure 6: Core performance trade-offs on the test set. The vertical axis represents health improvement relative to the historical baseline, while the horizontal axis shows the budget ratio compared to historical spending. The origin (0,1) represents the historical baseline performance and cost. The upper-left quadrant is the optimal region.

**multitask_cpq: The Theoretical Optimum.** As shown in Figure 6, multitask_cpq dominates the optimal upper-left quadrant. It achieves a remarkable 28.30% health improvement over the historical baseline while theoretically requiring only 33.7% of the budget. Its efficiency is unparalleled at 12.88 health units per $1M. This superior performance stems from its discovery of a highly effective, low-cost maintenance strategy (favoring Action 1) that deviates significantly from expert behavior (behavioral similarity of only 0.139). However, its primary weakness is its complete disregard for budgetary constraints, evidenced by a 71.8% violation rate. Therefore, its position on the plot represents an idealized, operationally infeasible outcome. Its value is not as a deployable agent, but as a benchmark for the maximum achievable performance if constraints could be perfectly managed.

**discrete_bc_50 and iqlcql_marl: The Pragmatic Compromises.** These two multi-agent algorithms are clustered near the historical baseline point (0,1). Discrete_bc_50 resides in the upper-right quadrant, delivering a solid 4.77% health improvement at a slightly increased budget ratio of 1.06. IQLCQL_MARL sits almost directly on the baseline, offering a marginal 1.12% health improvement for a slightly reduced budget ratio of 0.97. Their key strength is their high fidelity to expert decision-making, with behavioral similarity scores of 0.943 and 0.944, respectively. This mimicry allows them to inherit the experts' implicit constraint adherence, resulting in manageable violation rates (around 8.6%). Their weakness is the flip side of this strength: by closely following historical patterns, they fail to discover novel, more efficient policies, limiting their performance gains. They represent the most reliable and deployment-ready options, offering modest but safe improvements.

**cdt: The Frugal Underperformer.** The Conditional Decision Transformer (cdt) is positioned in the lower-left quadrant. Its primary strength is frugality, achieving a low budget ratio of 0.348.

However, this cost-saving comes at the expense of performance, resulting in a 4.71% degradation in network health compared to the historical baseline. While it exhibits high behavioral similarity (0.859), it appears to have learned a policy that is overly conservative, prioritizing inaction or the lowest-cost options to a degree that is detrimental to the long-term health of the infrastructure. The trade-off it makes—sacrificing health for cost savings—is ultimately suboptimal.

**`qmix_cql` and `multitask_bc`: The Inefficient Spenders.** Both of these algorithms are located in the undesirable lower-right quadrant, indicating that they spend more money to achieve worse results than the historical baseline. `QMIX-CQL` incurs a 75% increase in budget (1.75 ratio) for a 16.62% decline in health. `Multitask_bc` is even less effective, with a 54% budget increase leading to a 40.95% health decline. Their clear weakness is a failure to learn a coherent policy from the offline data. For `qmix_cql`, this likely points to the inherent difficulty of learning a joint multi-agent value function from a static dataset. For the single-agent `multitask_bc`, it suggests that a centralized agent cannot effectively manage the spatially and temporally complex decisions required for the entire network, leading to inefficient and poorly coordinated actions. These algorithms serve as important baselines, demonstrating that naive application of standard methods can be highly detrimental.

### C.2 How is the algorithm's sensitivity to budget?

To investigate how algorithms adapt their policies in response to varying levels of resource availability, we conducted a 100-year longitudinal simulation, scaling the total budget from $0.25\times$ to $4.0\times$ the historical average. Figure 7 illustrates the response of six key algorithms, revealing two distinct behavioral archetypes.

**Budget-Sensitive Algorithms.** The first three algorithms in the figure—`discrete_bc_50`, `multitask_bc`, and `iqlcql_marl`—demonstrate a clear and predictable response to budget changes. As the budget multiplier increases, their total expenditure (blue line) rises almost linearly. Correspondingly, the average network health (green line) also improves, confirming that these algorithms effectively utilize additional resources to perform more or higher-cost maintenance. However, the efficiency (orange line), representing health gain per dollar, shows a clear trend of diminishing returns; doubling the budget does not double the health outcome. This behavior confirms they have learned a direct association between resource levels and action selection from the training data.

**Budget-Insensitive Algorithms.** In contrast, the latter three algorithms—`cdt`, `multitask_cpq`, and `qmix_cql`—exhibit more irregular patterns. For `multitask_cpq`, increasing the budget has a negligible effect on its expenditure, which remains consistently low. Its policy is rigidly governed by its learned Q-values and constraint model, not by the opportunity to spend more. `CDT` and `QMIX-CQL` show non-monotonic and unpredictable spending habits. This behavior is attributed to their underlying optimization objectives, which are not designed to simply consume a budget but to achieve other goals (e.g., a target return for CDT). This irregular behavior is likely exacerbated by distribution shift, as large deviations from the training data's budget conditions create out-of-distribution states that challenge the models' generalization capabilities.

### C.3 How does the original dataset's allocation strategy compare to rule-based alternatives?

To investigate the impact of resource distribution, we tested several rule-based allocation strategies against the empirically derived strategy from the historical data. To clarify the methodologies tested, Table 7 provides a comprehensive description of the main budget allocation strategies evaluated in our 100-year simulations.

For this analysis, we focused exclusively on the three algorithms identified as 'budget-sensitive'—`multitask_bc`, `iqlcql_marl`, and `discrete_bc_50`. This selection is deliberate: since budget-insensitive algorithms do not consistently respond to changes in the *total* budget, they are unlikely to yield meaningful insights into the nuances of budget *distribution*. An experiment on allocation strategy is only meaningful for models that have learned to actively manage a given budget. Figure 8 presents the outcomes for these three algorithms under the different allocation strategies.

Table 7: Description of Budget Allocation Strategies.

| Strategy Name | Description | Allocation Rule ($B_i$ is budget for bridge $i$) |
|---|---|---|
| *Basic and Historical Strategies* | | |
| **Original** | Mimics the historical spending distribution from the dataset. | $B_i = B_{total} \cdot \frac{C_i^{historical}}{\sum_j C_j^{historical}}$ |
| **Uniform** | Distributes the total budget equally among all bridges. | $B_i = \frac{B_{total}}{N}$ |
| **Uniform Top-10% Importance** | Concentrates the budget equally on the 10% of bridges with the highest importance scores. | $B_i = \begin{cases} \frac{B_{total}}{0.1N} & \text{if } i \in \text{Top 10\% Importance} \\ 0 & \text{otherwise} \end{cases}$ |
| *Health-Driven Strategies* | | |
| **Critical First** | Allocates budget only to bridges in a critical health state ($H_i \le 0.3$). | $B_i = \begin{cases} \frac{B_{total}}{|\{j:H_j \le 0.3\}|} & \text{if } H_i \le 0.3 \\ 0 & \text{otherwise} \end{cases}$ |
| **Health Threshold** | Generalizes 'Critical First', allocating budget to bridges below a health threshold $\theta$. | $B_i = \begin{cases} \frac{B_{total}}{|\{j:H_j < \theta\}|} & \text{if } H_i < \theta \\ 0 & \text{otherwise} \end{cases}$ |
| **Preventive** | Focuses budget on bridges in a moderate health range to prevent deterioration. | $B_i = \begin{cases} \frac{B_{total}}{|\{j:0.3 \le H_j \le 0.7\}|} & \text{if } 0.3 \le H_j \le 0.7 \\ 0 & \text{otherwise} \end{cases}$ |
| *Integrated and Cyclical Strategies* | | |
| **Importance-Health Weighted** | Allocates budget proportionally to a weighted score of importance and health. | $\text{Score}_i = 0.6 \cdot \text{Imp}_i + 0.4 \cdot (1 - H_i); B_i = B_{total} \cdot \frac{\text{Score}_i}{\sum_j \text{Score}_j}$ |
| **Rotating Focus** | Divides bridges into 3 groups, cycling the full budget between them yearly. | $B_i = \begin{cases} \frac{B_{total}}{N/3} & \text{if } i \in \text{active group for year } t \\ 0 & \text{otherwise} \end{cases}$ |

A striking observation from Figure 8 is the remarkable consistency in the relative performance of the strategies across all three algorithms. The **original allocation strategy** from the dataset consistently achieves a superior balance, positioning it in the upper-right region of the Health vs. Cost plots. This indicates it secures high health outcomes for a moderate and efficient total cost.

In contrast, rule-based heuristics demonstrate clear deficiencies. Strategies that focus resources on a small subset of bridges (e.g., `uniform top-10% important`) result in high efficiency but lead to poor overall network health because a large portion of the infrastructure is neglected. Conversely, strategies that distribute the budget uniformly (`uniform`) lead to excessive expenditure. This is because they create a dense budget allocation that is far out-of-distribution from the sparse allocations seen during training, causing the algorithms to utilize resources inefficiently. This analysis validates the expert-driven allocation strategy found in the historical data as a robust and well-balanced heuristic and underscores the critical importance of maintaining consistency between the distributional characteristics of the training data and the allocation method used during deployment.

## D  HYPERPARAMETER AND COMPUTATIONAL DETAILS

This section provides a comprehensive summary of the hyperparameter configurations, training procedures, and computational environment used for all experiments. The parameters were determined through a combination of standard practices in offline reinforcement learning literature and a limited grid search. **Table 8** outlines the general optimization, network, and batching settings that were applied across most algorithms. Following this, **Table 9** delves into the unique architectural and algorithmic parameters specific to each model family, such as those for Decision Transformers and Conservative Q-Learning. Finally, **Table 10** details the hardware and software stack used for the experiments, along with the resulting training performance metrics like duration and memory usage.

## E  MORE DISCUSSION

This section provides additional details and nuances that complement the main discussion in the body of the paper. We delve deeper into the core findings, the specific benefits and drawbacks of multi-agent formulations, and the inherent limitations of this study.

### E.1  THE CONSTRAINT-PERFORMANCE PARADOX AND POLICY DISCOVERY

A central finding of our study is a fundamental paradox: algorithms optimized for maximal performance gains consistently exhibit the poorest adherence to operational constraints. `Multitask CPQ` epitomizes this conflict. Its superior performance, marked by a 38.03% health improvement over the historical baseline, is not arbitrary. This gain is attributed to the algorithm's identification of a novel and highly efficient policy—a strong preference for "Action 1," which represents low-cost, preventative maintenance. This discovery highlights the potential of reinforcement learning to identify superior policies that deviate from established human practice, in this case suggesting the

Table 8: General Training Hyperparameters. These settings were applied across all applicable algorithms unless specified otherwise.

| Parameter | Value / Setting |
|---|---|
| *Optimization* | |
| Learning Rate (Policy/Actor) | 3e-4 |
| Learning Rate (Value/Critic) | 1e-3 |
| Optimizer | Adam |
| Adam Betas $(\beta_1, \beta_2)$ | (0.9, 0.999) |
| Learning Rate Schedule | Cosine Annealing with Warm Restarts |
| Weight Decay | 1e-4 |
| Gradient Clipping Norm | 10.0 |
| *Network Architecture* | |
| Hidden Layer Dimensions | 64-128 |
| Network Depth | 2-3 hidden layers |
| Activation Function | ReLU |
| Dropout Rate | 0.1 |
| *Batching and Training Loop* | |
| Batch Size (Single-Agent) | 1024 |
| Batch Size (Multi-Agent) | 16 episodes |
| Training Steps | 200,000 |
| Evaluation Frequency | Every 2 epochs |
| Early Stopping Patience | 20 evaluations |

Table 9: Algorithm-Specific Hyperparameters.

| Algorithm Family | Parameter | Value / Setting |
|---|---|---|
| **Decision Transformer (CDT)** | Context Length | 10 timesteps |
| | Embedding Dimension | 128 |
| | Transformer Layers | 4 |
| | Attention Heads | 8 |
| | Return-to-Go Normalization | Per-episode z-score normalization |
| | Layer Normalization | Applied |
| **Conservative Q-Learning (CQL)** | CQL Regularization Weight ($\alpha$) | Tuned in {2.0, 3.0} |
| *(Used in IQL-CQL, QMIX-CQL, CPQ)* | Discount Factor ($\gamma$) | 0.95 |
| | Target Network Update Rate ($\tau$) | 0.005 (soft update) |
| | Target Update Frequency | Every 100 training steps |
| | Temperature Parameter ($\beta$) | 1.0 |
| **Recurrent Networks (RNN)** | RNN Type | GRU |
| *(Used in MARL-BC, IQL-CQL, QMIX-CQL)* | RNN Hidden State Dimension | 64 |
| | Sequence Sampling Length | 32 |
| | Data Augmentation | Time-shift augmentation |
| **QMIX-CQL** | Mixer Network Hidden Dimension | 32 |
| | Centralized Training Sampling | Episode-based |

long-term value of frequent, minor interventions. However, the unconstrained pursuit of this strategy resulted in a 71.8% budget violation rate, rendering the policy operationally infeasible. In contrast, imitation-based approaches such as `discrete_bc` achieve more modest but reliable improvements by closely replicating the demonstrator's behavior, thereby inheriting its implicit adherence to constraints.

The 100-year simulations reveal another critical challenge: long-term sustainability and distribution shift. While many deployed policies can successfully mitigate network-wide deterioration over a century-long horizon, they often do so at the cost of substantially increased expenditures that may prove economically unsustainable. This issue is compounded by a distribution shift, wherein algorithms exhibit performance degradation when encountering out-of-distribution states not prevalent in the training data, such as a network of predominantly healthy bridges. The behavior of budget-sensitive algorithms further suggests that historical expert decisions reflect a calculated compromise

Table 10: Computational Environment and Training Performance.

| Component | Specification / Value |
|---|---|
| *Hardware Specifications* | |
| GPU | $10 \times$ NVIDIA GeForce RTX 2080Ti (11GB VRAM) |
| System RAM | 256 GB |
| Storage | High-speed SSD |
| *Software Dependencies* | |
| Framework | PyTorch 2.4.1, CUDA 13.0 |
| Multi-Agent Library | Custom implementation based on PyMARL |
| Data Processing | Pandas, NumPy, SciPy |
| *Training Performance Metrics* | |
| Total Training Time (All Algs) | $\sim$180 minutes |
|   - Single-Agent BC | 15-20 minutes |
|   - Multi-Agent BC | 25-30 minutes |
|   - CDT | 60-75 minutes |
|   - CPQ | 25-30 minutes |
|   - IQL-CQL | 10-15 minutes |
|   - QMIX-CQL | 20-25 minutes |
| Peak GPU Memory Usage | 8-10 GB per GPU for largest models |

between ideal maintenance outcomes and fiscal responsibility, rather than constituting a purely sub-optimal strategy.

For immediate, risk-averse applications, our results suggest that imitation-based methods offer the most pragmatic solution due to their reliable constraint adherence. These methods should therefore be viewed not as a definitive goal, but as a robust safety baseline. The primary promise of reinforcement learning lies in its capacity to surpass the limitations of historical data and discover novel, more efficient operational strategies, as evidenced by `multitask_cpq`. The imperative for future research, underscored by our benchmark, is to develop algorithms that reconcile this innovative potential with the strict constraint satisfaction mandated by safety-critical systems.

### E.2 MULTI-AGENT COORDINATION: BENEFITS AND LIMITATIONS

The adoption of a multi-agent framework yields distinct advantages when appropriately designed for an offline learning context. Multi-agent imitation learning algorithms, such as the discrete behavioral cloning variant, consistently outperformed their single-agent counterparts. This suggests they successfully capture the spatial interdependencies inherent in expert maintenance decisions, where the state of one asset influences actions taken on another. However, our results also highlight the challenges for value-based multi-agent methods. `QMIX-CQL`, for instance, struggled to derive a stable and effective policy. This difficulty likely stems from the fundamental challenge of estimating a joint multi-agent value function from a static, fixed dataset without the ability to perform environmental exploration. This underscores a significant limitation of current value-based multi-agent RL techniques in safety-critical offline settings, where inaccurate value estimates can lead to unpredictable and unreliable policies.

### E.3 LIMITATIONS OF THE CURRENT STUDY

This research is subject to several limitations that should be considered when interpreting the results and which present clear avenues for future work.

**Dataset and Expert Bias.** The National Bridge Inventory (NBI) dataset, while extensive, provides only 30 years of temporal data. This duration may not be sufficient to capture the complete lifecycle dynamics of infrastructure assets that have service lives exceeding 50 or 100 years. Furthermore, the expert decisions recorded in the dataset are not a pure reflection of optimal engineering practice; they are inherently biased by historical budget constraints, organizational policies, and established

conventions. Consequently, algorithms trained via imitation risk perpetuating these potentially sub-optimal patterns.

**Model and Simulation Simplifications.** The state transition model, although empirically derived from historical data, is necessarily a simplification of the complex, stochastic processes governing infrastructure deterioration. Our simulation environment abstracts away numerous real-world factors that can significantly impact maintenance planning and outcomes. These include, but are not limited to, the effects of severe weather events, the need for emergency repair interventions, traffic disruptions caused by maintenance activities, and the complex logistics of stakeholder coordination.

**Offline Learning Paradigm.** The offline setting, by definition, precludes algorithmic adaptation to evolving operational conditions or novel environmental stressors not represented in the training dataset. While our evaluation methodology is extensive, it ultimately relies on a simulated environment. Despite our efforts to ensure its realism, such a simulation cannot fully replicate the multifaceted complexity and unpredictability of real-world infrastructure management scenarios.

## F    STATEMENT ON THE USE OF AI TOOLS

In the preparation of this manuscript, we utilized an artificial intelligence (AI) writing assistant, specifically [e.g., gemini-2.5], to aid in the language polishing and proofreading process. The use of the AI tool was primarily for improving grammar, enhancing sentence fluency, and correcting spelling errors. The core content, ideas, and analysis presented in this paper are entirely the original work of the authors. We manually reviewed and confirmed all suggestions proposed by the AI and take full responsibility for the final content of the article.

## G    REPRODUCIBILITY AND CODE AVAILABILITY

### G.1    REPRODUCIBILITY MEASURES

**Deterministic Operations:**

- Fixed random seeds (42 ,1024, 2025 for training)
- Deterministic GPU operations where computationally feasible
- Consistent data loading order across experiments
- Fixed initialization schemes for all neural networks

**Environment Control:**

- Conda environment specifications with exact version pinning
- Detailed documentation of hardware configurations

### G.2    DATA AND MODEL AVAILABILITY

Upon paper acceptance, the following will be publicly available:

- **Preprocessed benchmark datasets** with comprehensive metadata
- **Trained model checkpoints** for all evaluated algorithms
- **Complete experimental configurations** with hyperparameter specifications
- **Evaluation scripts** for reproducing all reported results
- **Visualization tools** for generating paper figures and custom analyses

The benchmark will be maintained as an open-source project with:

- Regular updates incorporating new NBI data releases
- Community contributions for algorithm implementations
- Comprehensive testing suites for reliability assurance
- Documentation updates reflecting methodological advances

Figure 7: Budget scaling sensitivity analysis for six algorithms. Each row displays an algorithm's response to budget multipliers (0.25×, 0.5×, 1.0×, 2.0×, 4.0×) under the original allocation strategy. The plots show average health (green), total expenditure (blue), and efficiency (orange) over the 100-year simulation.

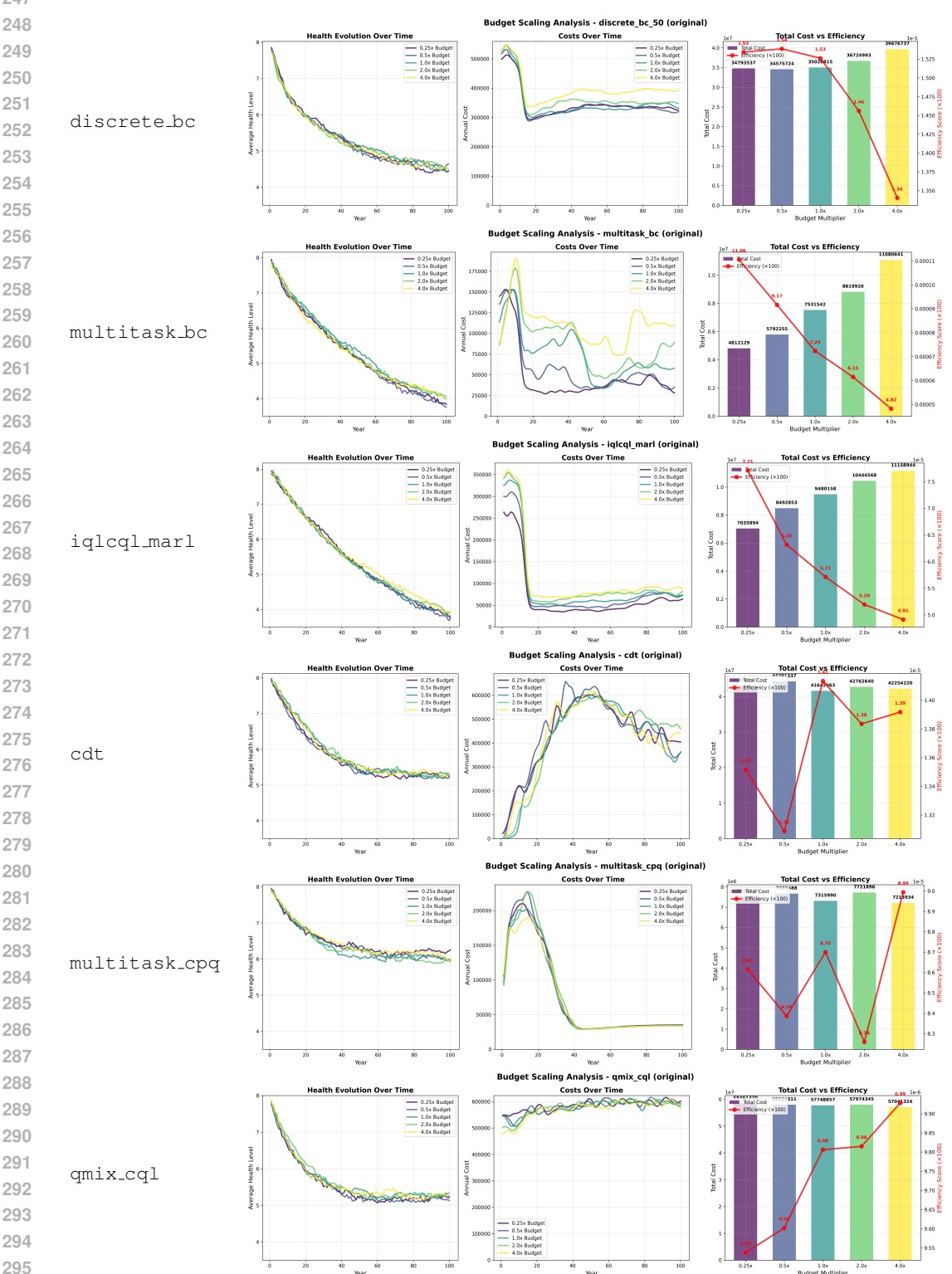

Figure 8: Comparison of different budget allocation strategies for budget-sensitive algorithms. Each row shows the performance of one algorithm under nine different allocation strategies, with total budget held constant. The plots show the trade-off between long-term health and total cost.

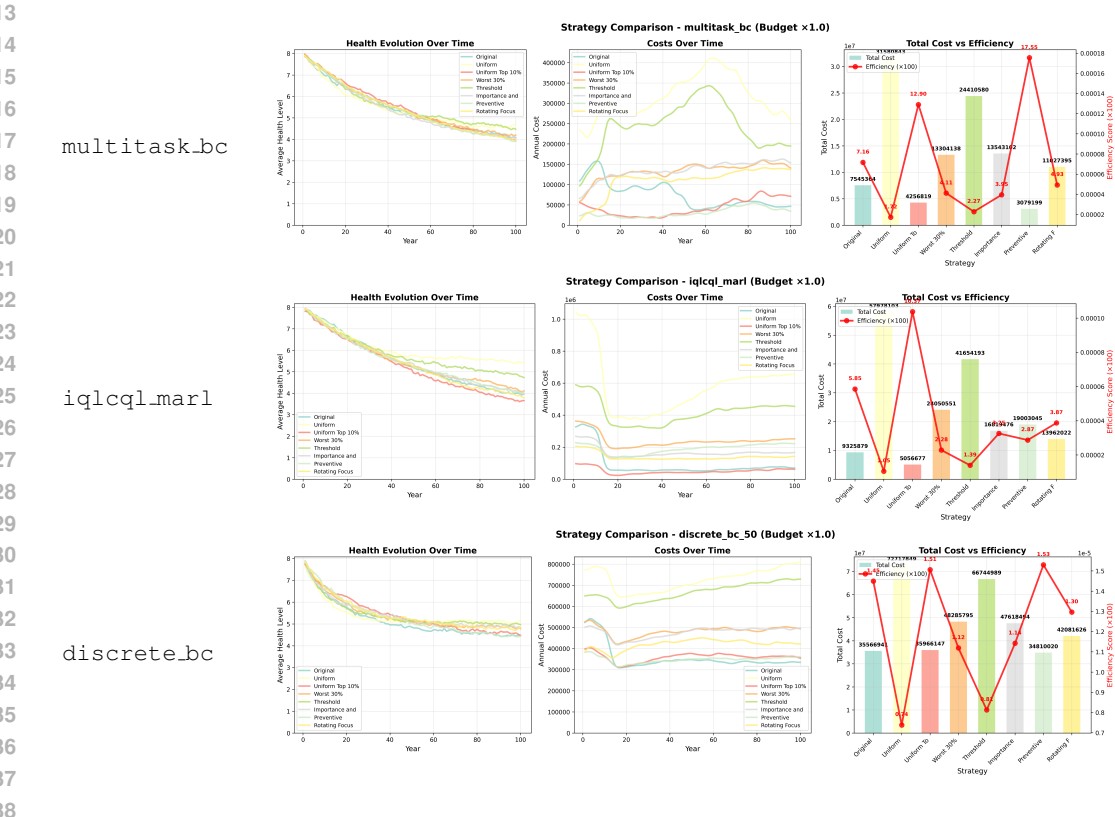