# OpenReview forum: "BRIDGEBENCH: AN OFFLINE CONSTRAINED MULTI- AGENT REINFORCEMENT LEARNING BENCHMARK FOR INFRASTRUCTURE MANAGEMENT"
_ICLR.cc/2026/Conference — ICLR 2026 Conference Withdrawn Submission_

### Official Review · Reviewer_KRzL · 2025-10-31

**Soundness:** 3
**Presentation:** 3
**Contribution:** 3
**Rating:** 6
**Confidence:** 4

**Summary:**

The paper discusses how reinforcement learning (RL) can be used to optimize bridge maintenance policies. Its core contribution is a framework for training RL models using existing datasets. The results indicate that RL can discover more efficient solutions than those based on expert policies, although such solutions may violate short-term budget constraints.

**Strengths:**

1. The paper is well written and easy to follow; it is a pleasant read.

2. It tackles an interesting interdisciplinary problem.

3. The results are encouraging, showing that substantial benefits can be achieved with RL. The most interesting scheme, in my view, is the single-agent multi-task_cpq, which favors early interventions to avoid later costly maintenance. However, this solution violates certain short-term budget limitations.

**Weaknesses:**

1. The technical contribution is limited. The primary contribution is a framework that enables training RL models from historical datasets. Thus, it is a solid applied RL paper, but it does not necessarily advance the algorithmic aspects of the studied RL methods.

2. Several parameters are unspecified. In particular, the “hard budget constraints” mentioned at the end of Section 3.1 are not clearly defined. What exactly are these constraints? Who defines them? How are they related to B? Shouldn’t the reward include these budget constraints (at least as cost terms)? It is difficult to follow the discussion of violation rates—and to understand why, e.g., multi-task_cpq is infeasible—without knowing the allocated budget for a given decision.

**Questions:**

1. Why is high behavior similarity preferred? Shouldn’t budget ratio, improvement, and health gain be prioritized over behavior similarity?

2. Would it be possible to apply an off-policy learning approach, where the algorithm learns from experts’ actions? Such a method might provide a better trade-off between behavior similarity and the other metrics.

3. Could hard budget constraints also be learned? The results suggest that these constraints increase total expenditure, as more budget-efficient solutions tend to violate them more often.

Please also see the questions raised in the Weaknesses section.

---

### Official Review · Reviewer_Fxkx · 2025-11-01

**Soundness:** 2
**Presentation:** 2
**Contribution:** 2
**Rating:** 2
**Confidence:** 4

**Summary:**

This paper introduces BridgeBench, a benchmark for offline constrained multi-agent reinforcement learning (MARL) using data from the U.S. National Bridge Inventory (NBI). The benchmark aims to support research in large-scale infrastructure management, specifically, bridge maintenance planning under budget constraints. It models bridge deterioration and maintenance as a multi-agent constrained Markov decision process (MA-CMDP) and evaluates offline RL and MARL algorithms (e.g., IQL-CQL, QMIX-CQL, CPQ, CDT, BC). The authors provide a processed dataset of ~2000 episodes, an empirical transition model, and simulations exploring long-term policy performance and budget sensitivity.

The topic is timely and relevant for applying RL in safety-critical domains, and the paper offers a comprehensive dataset and experimental analysis. However, methodological clarity, evaluation rigor, and the framing relative to prior work require substantial revision to meet ICLR standards.

**Strengths:**

+ Timely and impactful problem: it addresses infrastructure maintenance under system-level budget constraints, a practically important but understudied area in the RL community.

+ Real-world data: deterioration, repair, and cost data are defined based on long-term NBI bridge records (1992–2023).

+ Comprehensive modeling pipeline, including preprocessing, feature engineering, transition-matrix estimation, and regional partitioning.

+ Discussion of constraint-performance trade-offs, identifying that unconstrained policies (e.g., CPQ) can achieve large gains but violate operational limits, highlighting an important research challenge.

+ Supplementary material provides transparency on data preparation and hyperparameters.

**Weaknesses:**

W1. Offline-learning assumptions
+ The dataset comprises approximately 2,000 episodes from empirical transitions, but the (long-term) dynamics are stationary and derived from a region (California). It is unclear whether this supports meaningful offline generalization beyond narrow data distributions. The feasibility of deploying trained offline RL models without a simulator or forward model is questionable; engineering simulators are usually required and commonly used for reliable evaluation.

W2. Positioning and novelty
+ The framing as a “first benchmark” is overstated. The contribution should be positioned with respect to existing works in infrastructure management planning benchmarks (e.g., IMP-MARL, NeurIPS 2023) and related works published in the engineering community (Andriotis, C. P. & Papakonstantinou, K. G. (2021). Deep reinforcement learning driven inspection and maintenance planning under incomplete information and constraints. Reliability Engineering & System Safety, 212, 107551; Saifullah, M., Papakonstantinou, K. G., Andriotis, C. P. & Stoffels, S. M. (2024). Multi-agent deep reinforcement learning with centralized training and decentralized execution for transportation infrastructure management. arXiv preprint arXiv:2401.12455.

W3. Problem formulation
+ Although the introduction refers to hard budget constraints, the mathematical formulation uses an expectation-based soft constraint. If this is intended, Lagrangian or primal-dual approaches should be compared; otherwise, the constraint handling is inconsistent with the stated goal.
+ No constrained RL/MARL baselines (e.g., Lagrangian IQL) are implemented, weakening the “constrained” aspect of the study.

W4. Reward definition
+ The reward includes arbitrary bonus/penalty terms and a cost weight β chosen heuristically ($\beta$ = 10⁻³). A more principled formulation linking rewards directly to state deterioration or failure costs would improve interpretability and applicability.
Reported cost magnitudes (e.g., $2,433 for replacement) appear unrealistic even if scaled; ratios between repair and replacement costs (< 2×) seem inconsistent with real bridge management economics.

W5. Observations and features
+ Ambiguity exists between the description of the state (line 168: multiple structural and traffic features) and the final 5-dimensional observation vector (line 226).

W6. Missing detail on N
+ The paper does not specify how many bridges (agents) each region contains or the total number of bridges included in the dataset. Since N determines the size of each multi-agent problem and affects algorithm scalability and coordination behavior, this omission makes it difficult to assess experimental complexity or to reproduce the benchmark.

W7. Evaluation methodology
+ Results lack training curves, confidence intervals, and seed counts. Statistical significance and variance across runs are not discussed.
Many figures (especially in the appendix) are of low resolution; some references to algorithms and other sources are uncited.
The claim of “100-year simulations” uses a fixed empirical transition model, which may not reflect long-term non-stationary deterioration.
+ The evaluation relies on custom, domain-specific metrics without standard RL baselines (e.g., average return, as mentioned previously), hindering comparability to prior work and to general offline RL benchmarks.

W8. Constraint enforcement
+ Algorithms are trained without constraints and only evaluated post-hoc with budget checks. This explains high violation rates and weakens conclusions about constraint adherence.

W9. Reproducibility
+ Code and processed datasets are not publicly available at submission, preventing independent validation. The work positions itself as a benchmark, but the experimental evaluation dominates the paper. Without public release of code, dataset splits, or environment interfaces, the contribution reads more as a single-study experiment than a reusable benchmark.

**Questions:**

Q1. Algorithm comparison: Why does IQL outperform QMIX despite lacking explicit coordination?

Q2. Cooperative vs competitive setting: The text alternates between cooperative and competitive interpretations of agents sharing a budget. Clarify the intended structure.

Q3. Connectivity matrix W: It is mentioned but not analyzed. Does W influence coordination, or is it unused beyond regional grouping?

Q4. Policy inspection: Qualitative visualization of learned maintenance patterns (e.g., sample regional rollouts) would help interpret behavior beyond average metrics. Could you provide illustrative policy realizations in the paper?

---

### Official Review · Reviewer_vMGz · 2025-11-01

**Soundness:** 1
**Presentation:** 1
**Contribution:** 2
**Rating:** 2
**Confidence:** 3

**Summary:**

The authors propose BridgeBench, a benchmark for bridge maintenance. It focuses on offline, constrained, multi-agent RL methods. Different methods are tested on it.

**Strengths:**

* It addresses an important yet sometimes overlooked real-world problem.
* Analysis of baselines on their benchmark.

**Weaknesses:**

* Offline RL, MARL and CRL are already very challenging in their own right. Offline RL in particular can only work if data coverage is strong enough, otherwise, it does not work. Comparing offline RL methods on its own is already difficult and unreliable, and we still do not fully understand how to build a good dataset. There are even papers trying to address this, e.g. "Understanding the Effects of Dataset Characteristics on Offline Reinforcement Learning" by Schweighofer et al.

* Including MARL and CRL will require even more data, as constraints must be supported by data and there must be enough data for every agent. This makes me highly suspicious of the applicability and usability of this benchmark. Current benchmarks only focus on one element to study it more carefully without complicating the problem further. There may be a need for this at some point in order to port RL to real-world domains, but I think we still need more analysis in each of these areas.

* Data-driven world models are used, however, these models can produce unrealistic data.

* When algorithms are used for evaluation, they are not properly cited, only the abbreviation is used, e.g. Conservative Q-Learning (CQL) (Kumar et al., 2020), or Implicit Q-Learning (IQL) (Kostrikov et al., 2021). I do not understand the baselines, such as CQL-IQL.

* Figure 3/4: The text is too small to read.

* Inconsistent writing style, e.g. Reinforcement Learning is written in upper case in the introduction, but in lower case in the conclusion. A lot of bold text is used in the introduction, which hinders the reading flow.

**Questions:**

* What is CQL-IQL? In Section 4.1, it is introduced as a state-of-the-art offline MARL method. However, to me, these are two offline RL methods. The same applies to QMIX-CQL.
* Could you tell me more about the world model for offline policy evaluation? This is a crucial part of the benchmark and its not covered in the main part of the paper.
* Is there only one task in this benchmark with different settings, or are many different tasks planned?

---

### Official Review · Reviewer_bqV1 · 2025-11-05

**Soundness:** 2
**Presentation:** 2
**Contribution:** 1
**Rating:** 2
**Confidence:** 4

**Summary:**

This paper introduces BridgeBench, a multi-agent constrained Markov decision process meant to simulate the problem of repairing bridges across a region subject to budgetary constraints. In BridgeBench, each bridge is represented as an individual agent, with observations representing the health and characteristics of the bridge, and 4 discrete actions ranging from “no action” to “replacement.” The goal of the agents is to maximize a global, shared reward function which accounts for changes in bridge health as well as the cost of actions taken. BridgeBench is designed for offline RL training, providing a historical dataset of bridge maintenance records from the National Bridge Inventory (NBI) for highway bridges located in California between 1992 to 2023. The paper evaluates 7 or 8 RL algorithms, finding trade-offs between RL algorithms that maximize bridge health improvements vs. RL algorithms that adhere to budgetary constraints.

**Strengths:**

**S1) Novel multi-agent RL benchmark based on real-world data**

BridgeBench introduces a novel multi-agent RL benchmark based on real-world data. While existing works have tested RL for infrastructure planning, BridgeBench appears to be novel for its use of offline data, according to the authors.

**S2) Experiments seem reasonably broad in coverage**

The paper tests a number of single- and multi-agent RL algorithms on BridgeBench.

**Weaknesses:**

**W1) Relevance of offline multi-agent RL for bridge problem**

My main concern is that I do not find that the authors have sufficiently justified why bridge maintenance is best formulated as a offline multi-agent RL problem.

First, why is RL necessary for infrastructure planning? Are traditional optimization-based strategies insufficient, e.g., stochastic optimization? It would be helpful to justify the RL use case if the authors can demonstrably show significant gains from using RL over non-RL baselines.

Second, why is it necessary to use multi-agent RL, as opposed to single-agent RL? In practice, infrastructure planning decisions are generally coordinated centrally by some government agency, such as the California Department of Transportation. Indeed, even the experiments seem to suggest that single-agent RL performs on-par if not better than the multi-agent RL algorithms. Because the setting is cooperative, and in practice there is a centralized coordinator, I do not find the use case of multi-agent RL convincing.

Finally, why only consider offline RL? I understand the high-level motivation for leveraging an offline dataset based on actual historical NBI records. However, the authors also estimate a transition kernel for the MDP, which then enables online RL. Therefore, why not also test online RL algorithms? It seems to me that the requirement of offline RL here is somewhat artificial.

**W2) Lack of clarity about evaluations**

I find the experimental evaluations to be confusing. I understand that there are two scenarios: static held-out test set, and 100-year longitudinal simulation. I have concerns about both.

For the static held-out test set, more details should be provided on how the evaluation is performed. How was the held-out test set chosen? Is there any overlap in the bridges from the training set and the held-out test set? How large was this held-out test set? Also, how do you even evaluate RL algorithms on a held-out test set, when the actions chosen by the RL algorithms differ from the historical data?

For the 100-year longitudinal simulation, more details should be provided on both the training and evaluation of the RL algorithms. Were the RL algorithms only trained on offline data, or were they allowed to update online over the course of the simulation? How many random seeds were used?

Finally, how were the RL algorithms chosen? I understand that descriptions of each algorithm are given in the supplementary materials, but a discussion about why these specific algorithms were benchmarked should be included. Also, please give appropriate citations to each algorithm.

**W3) Concerns about MDP design**

The design of the MDP for the bridge maintenance problem is interesting, but some key details and design choices are either missing or concerning to me.

For example, the MDP is missing details such as:

- what is the dimension of the state space?

- does every agent observe the same state space, or only state corresponding to a single bridge?

- what is the number of agents/bridges $N$ used in the experiments?

- what is the length of the horizon $T$ used in the experiments?

- what is the role of the connectivity matrix $W$?


Also, I am concerned by how the empirical state transition matrix is constructed. Figure 5 is meant to show the state transition matrix, but the vast majority of the probability mass is concentrated on the diagonal, regardless of the action taken. This seems counter-intuitive to me. I would expect, for example, bridge replacement to almost always convert bridge health into the “Good” category, but this does not seem to be the case.

**W4) Missing discussions on relevant related works and domains**

First, I find it surprising that the paper makes no mention of the off-policy RL literature, which is very closely tied to the offline RL literature. See, e.g., the following papers:

[1] Offline RL Without Off-Policy Evaluation.
David Brandfonbrener, Will Whitney, Rajesh Ranganath, Joan Bruna
NeurIPS 2021
[https://proceedings.neurips.cc/paper_files/paper/2021/hash/274a10ffa06e434f2a94df765cac6bf4-Abstract.html](https://proceedings.neurips.cc/paper_files/paper/2021/hash/274a10ffa06e434f2a94df765cac6bf4-Abstract.html)

[2] Empirical Study of Off-Policy Policy Evaluation for Reinforcement Learning
Cameron Voloshin, Hoang Le, Nan Jiang, Yisong Yue
NeurIPS 2021
[https://datasets-benchmarks-proceedings.neurips.cc/paper/2021/hash/a5e00132373a7031000fd987a3c9f87b-Abstract-round1.html](https://datasets-benchmarks-proceedings.neurips.cc/paper/2021/hash/a5e00132373a7031000fd987a3c9f87b-Abstract-round1.html)

Second, the paper should also discuss some related RL benchmarks for infrastructure such as SustainGym. For example, while SustainGym is not focused on offline RL, it does use historical data as part of its simulations.

[3] SustainGym: A Benchmark Suite of Reinforcement Learning for Sustainability Applications
C. Yeh, V. Li, R. Datta, J. Arroyo, N. Christianson, C. Zhang, Y. Chen, M. Hosseini, A. Golmohammadi, Y. Shi, Y. Yue, and A. Wierman
NeurIPS 2023
[https://papers.nips.cc/paper_files/paper/2023/hash/ba74855789913e5ed36f87288af79e5b-Abstract-Datasets_and_Benchmarks.html](https://papers.nips.cc/paper_files/paper/2023/hash/ba74855789913e5ed36f87288af79e5b-Abstract-Datasets_and_Benchmarks.html)

**W5) Concerns about Reproducibility**

I am mildly concerned that there is no link to an anonymous code repo, nor any code provided in the supplementary files. If the authors intend for BridgeBench to be a usable benchmark, then the authors will need to demonstrate some effort into making the benchmark usable by others.

**Questions:**

Q1) How does the performance of the RL algorithms tested on BridgeBench compare to their performance on other RL benchmarks?

---

### Note · Authors · 2025-12-26

**Comment:**

To the Area Chair and Reviewers,

After carefully considering the reviews and the constructive feedback provided, we have decided to withdraw our submission.

We are sincerely grateful for the time and effort the reviewers dedicated to evaluating our work. We find the comments extremely insightful and helpful. We fully agree with the concerns raised by the reviewers, particularly regarding the comprehensiveness of related work, the comparison with non-RL baselines, the clarity of our experimental settings, and the necessity of open-sourcing our code.

We realize that our paper requires significant revision to address these issues and meet the high standards of this conference. We plan to use this opportunity to substantially improve our work by incorporating your suggestions and refining the manuscript. We are also committed to organizing our code for release to ensure reproducibility in our future submission.

Thank you again for your valuable guidance and service to the community.

Sincerely,

The Authors

**Withdrawal Confirmation:**

I have read and agree with the venue's withdrawal policy on behalf of myself and my co-authors.